# Spatio-Temporal Patterns of County Population Shrinkage and Influencing Factors in the North–South Transitional Zone of China

**DOI:** 10.3390/ijerph192315801

**Published:** 2022-11-27

**Authors:** Tong Wu, Beibei Ma, Yongyong Song

**Affiliations:** School of Geography and Tourism, Shaanxi Normal University, Xi’an 710119, China

**Keywords:** population shrinkage, north–south transitional zone, east–west corridor effect, GeoDetector

## Abstract

Population is the foundation of socio-economic development. However, continued population shrinkage has made the problem of unbalanced and insufficient regional development more prominent, threatening human well-being. How to solve the contradiction between population shrinkage and regional development has become an urgent scientific problem. Therefore, taking a typical underdeveloped mountainous region, the North–South Transitional Zone of China, as an example, we analyzed the spatial and temporal evolution of regional population shrinkage from 2000 to 2020, classified the types of regional population shrinkage, and revealed the key influencing factors and driving mechanisms for the formation of population shrinkage patterns in poor mountainous counties. The results showed that: (1) From 2000 to 2020, the number of counties in the North–South Transitional Zone of China with population shrinkage grew, and the degree of shrinkage increased. The shrinking counties were mainly municipal counties, and the shrinkage types were mainly continuous shrinkage and expansion followed by shrinkage. (2) Spatially, the shrinking counties had significant and strengthening spatial autocorrelation, with obvious characteristics of the contiguous shrinkage of county units, and the shrinkage center of gravity and shrinkage agglomeration areas showed an evolutionary trend of shifting from east to west. The shrinking counties had obvious divergence in both the “east–west” and “north–south” directions. (3) Natural factors had an endogenous rooting role, while human factors had a strong driving role, and the impact of different influencing factors varied significantly. (4) The formation and evolution of the spatial pattern of county population shrinkage was subject to the synergistic effect of natural factors and human factors. The interaction between natural and human factors had a non-linear enhancement effect and a two-factor enhancement effect. The results of this study are expected to provide a scientific basis for coordinating regional human–land relations in order to optimize population-flow governance and sustainable regional development in the North–South Transitional Zone and less-developed regions of China.

## 1. Introduction

Currently, the phenomenon of population shrinkage is expanding worldwide, with the degree of shrinkage continuing to intensify, and population shrinkage has become an important issue facing global sustainable development [1,2,3,4]. Along with economic transformation and industrial restructuring, a declining fertility rate and aging among China’s population have gradually emerged since the start of the 21st century [5], and the problem of population shrinkage has become a major challenge for regional population security and economic and social development. People are the drivers of all economic activities [6], and population shrinkage will have a widespread and interlocking impact on the sustainable development of regions in several regards, such as economic, social, and environmental-resource-related [7]. In the context of the modern transformation of population growth patterns, the scientific understanding of the spatial differences in population shrinkage, its occurrence mechanisms, and its socio-economic effects is crucial for improving the resilience of regional development, and it is also an important theoretical and practical issue that needs to be studied in the field of population migration and population urbanization in contemporary China.

Since the start of the 21st century, population shrinkage has become a hotspot of widespread interest in the fields of sociology, economics, and geography and in various government organizations [8,9,10]. Studies by foreign scholars have focused on the changing trends [11,12], measurement methods [13], spatial divergence characteristics [14,15] and influencing factors [16,17] of population shrinkage. Numerous empirical studies have shown that population shrinkage is more severe in economically disadvantaged countries and less-developed regions compared to developed countries and regions with persistently low fertility. Differences in the level of economic development between regions are considered to be the main factor leading to population migration from less-developed to developed regions. Meanwhile, the cyclical cumulative effects of the natural environment, ethnicity, policies, disasters, and diseases along with economic factors further exacerbate the population shrinkage in less-developed regions [18,19,20,21]. Compared with the West, the population shrinkage phenomenon emerged relatively late in China. In 2010, geographers in China began to pay attention to the spatial characteristics and patterns of the population shrinkage phenomenon. In terms of spatial differentiation, at the regional scale, population shrinkage areas are not only distributed in less-developed regions such as the northeast and the central and western regions, but also have a more obvious distribution in developed coastal regions such as Beijing–Tianjin–Hebei, the Yangtze River Delta, and the Pearl River Delta [22], and show an evolutionary trend of continuous expansion [23]. At the county scale, population shrinkage is widespread in China [22], and county-level administrative units with persistently decreasing population densities are mainly distributed in China’s traditional population-loss areas, namely Hubei, Anhui, Henan, and other central regions [24]. In terms of influencing factors, existing studies generally agree that the migration flow of the population during rapid urbanization is the main cause of spatial differences in population expansion or shrinkage among regions [25,26]. A low level of economic development is the dominant factor in the formation of population shrinkage areas [4,27]. Furthermore, a low level of industrial development, an aging population, and a low level of urbanization are the main causes of population shrinkage [28,29], with a decline in natural population growth also representing an important factor [30,31], and a low level of infrastructure and public services having a further negative impact on regional population change [32,33]. Other studies have noted that natural factors also have some degree of influence on regional population shrinkage with significant geographical differences; for example, population shrinkage in Kitakyushu, Japan, is influenced by its high altitude and steep terrain [34], and population shrinkage in the China’s northeast areas is influenced by the cold climate [35,36]. However, in general, natural factors have a weaker influence on population shrinkage than socio-economic factors.

Population shrinkage is a regionally integrated phenomenon, and the processes and driving mechanisms of its various environmental and socio-economic influencing factors are highly complex and uncertain, displaying significant scale effects and geographical differences [37]. Although existing studies have discussed the spatial patterns and influencing factors of population shrinkage, in order to clarify the spatial characteristics and evolutionary mechanisms of the population shrinkage phenomenon during period of socio-economic transformation and rapid urbanization in China over the past 20 years, a richer and more geographically specific case area is needed for empirical analysis and theoretical framework construction through systematic studies of long time series. As the most important “geography–ecology” transitional zone in mainland China, the North–South Transitional Zone has a significant central location and high circulation characteristics. At the same time, the predominantly mountainous landscape brings about spatial heterogeneity in the natural environmental background. In terms of regional socio-economic development, the region is a typical underdeveloped inland mountainous area, one of the largest contiguous special hardship areas in China, and one of the main implementation areas for China’s ecological migration policy. During the period of rapid urbanization that has lasted more than 40 years, the population loss in the North–South Transitional Zone of China has been remarkable, and the region has become a concentrated distribution area of population shrinkage [22]. Therefore, we chose the North–South Transitional Zone of China as our study area to explore the spatial and temporal characteristics and regional differences of the population shrinkage phenomenon in less-developed mountainous areas with high circulation at the county scale based on the data of three population censuses (2000, 2010, and 2020), and we revealed the driving mechanisms behind the spatial evolution of population shrinkage in the study area from the perspective of the interaction between natural and human factors. The results of the study may help to enrich the theoretical and practical research on population shrinkage in less-developed mountainous areas during this transition period of regional population growth in China and provide a scientific reference for the governance of population mobility and sustainable regional development in China and other less-developed countries and regions around the world.

## 2. Materials and Methods

### 2.1. The Study Area

The North–South Transitional Zone of China is located between 102°54′–112°40′ E and 30°50′–34°59′ N, representing the largest ecological corridor on the Chinese mainland (Figure 1). It is also an important water conservation area for the South–North Water Diversion Central Project and comprises a vast region of underdeveloped mountainous landscape. The North–South Transitional Zone has become a region with a wide distribution of China’s current population shrinkage, and the long-term population outflow has led to a substantial lag in the socio-economic development level of the region [38]. According to the “National Science and Technology Basic Resources Survey Special Project: Comprehensive Scientific Investigation of the North–South Transitional Zone of China”, the whole region comprises six provinces and cities (namely Henan, Shaanxi, Gansu, Hubei, Chongqing, and Sichuan), covering 41 municipal districts, 24 county-level cities, 107 counties, 1 forest area, and 1 autonomous county, with an area of about 222,300 km^2^. The altitude of the region is between 13 and 5528 m; the terrain is high in the west and low in the east; and the topography is dominated by mountains, basins, and hills. In 2020, the total population of the North–South Transitional Zone of China was 71,442,800; its proportion of the country’s GDP was 3.84%; and the per capita GDP was 53,317 yuan, which was only 75.2% of the national average. Overall, the study area is characterized by significant variation in natural environmental background and socio-economic development, a weak industrial support capacity, prominent poverty, and intensive ecological construction projects.

### 2.2. Methodology

#### 2.2.1. Population Shrinkage Index

In this study, we used the population shrinkage index [39] to measure the degree of shrinkage in the resident population of the counties in the study area. Its formula was as follows:(1)$$C_{i}=\frac{P_{m}-P_{n}}{P_{n}}\times 100$$
where *C_i_* is the population shrinkage index; *P_m_* and *P_n_* are the number of permanent residents in *m* and *n* years; and *I* is a county-level administrative unit.

#### 2.2.2. Spatial Autocorrelation Analysis

Spatial autocorrelation analysis determines whether there is spatial correlation between an attribute value of an evaluated unit and the same attribute value of its neighboring areas in geographic space [40,41]. In this study, we used this method to explore whether there was spatial correlation among the population shrinkage throughout the study area, which was reflected by the global Moran index. Its formula was as follows:(2)$$I=\frac{\sum_{i=1}^{n}\sum_{j=1}^{n}w_{ij}\left(x_{i}-\bar{x}\right)\left(x_{j}-\bar{x}\right)}{\sigma^{2}\sum_{i=1}^{n}\sum_{j=1}^{n}w_{ij}}$$
where *x_i_* and *x_j_* are the attribute values of evaluation units *i* and *j* (*i* ≠ *j*); $\bar{x}$ is the average value of the attribute value of the evaluation unit; *n* is the number of counties; *I* is the global Moran index; *W_ij_* is a spatial weight matrix, which was defined by Rook adjacency in this paper; and *σ* is the sample variance. A significant positive value of *I* indicated a positive spatial autocorrelation among the population shrinkage at the county level, and vice versa.

#### 2.2.3. GeoDetector

GeoDetector is a set of statistical methods used to detect the spatial heterogeneity of a phenomenon in order to reveal the driving forces behind it [42]. In this study, factor detectors and interaction detectors were used to explore the effects of factors and their interactions on the spatial heterogeneity of county population shrinkage in the study area. The factor detector was mainly used to detect the spatial heterogeneity of the dependent variable *Y* and the ability of different influencing factors *X_i_* to explain the spatial heterogeneity of the dependent variable *Y*, and was measured by the *q* value. Its formula was as follows:(3)$$q=1-\frac{1}{Nr^{2}}\overset{k}{\underset{m-1}{\sum }}N_{m}r_{m}^{2}$$
where *K* is the partition of various influencing factors; *N* is the total number of units in the study area; *N_m_* is the number of units in different partitions of influencing factors; *r^2^* is the variance of *Y* value of all study samples across the whole study area; and $r_{m}^{2}$ is the sample variance within type *m* of influencing factor *X*. The value range of *q* was between 0 and 1. The closer the value of *q* was to 1, the more significant the impact of this factor was on population shrinkage.

The interaction detector was used to detect the interaction between different influence factors *X_i_*, that is, the interaction between two factors, and to detect whether the explanatory power of the dependent variable *Y* was enhanced or weakened under their combined action. The interaction results include five types of interactions (Table 1).

### 2.3. Data Sources

The resident population data by county used in this paper were obtained from the population censuses held in 2000, 2010, and 2020, which were carried out and published by the State Council of China and the National Bureau of Statistics (http://www.stats.gov.cn accessed on 6 October 2021) and represent the most authoritative and refined population survey data in China. DEM data were obtained from the Data Center for Resource and Environmental Sciences, Chinese Academy of Sciences (https://www.resdc.cn accessed on 15 October 2021). Elevation and topographic relief information was extracted from the DEM data in ArcGIS software. Socio-economic data were derived from the China County Statistical Yearbook and the statistical bulletin on national economic and social development for each county in the North–South Transitional Zone. The road-network data were obtained from the National Basic Geographic Information Center (http://www.ngcc.cn/ngcc accessed on 18 October 2021), with railroads, national roads, and provincial roads selected as the focus. The administrative boundary data of the North–South Transitional Zone were based on the “National Science and Technology Basic Resources Survey Special Project: Comprehensive Scientific Investigation of the North–South Transitional Zone of China”.

## 3. Results

### 3.1. Spatio-Temporal Evolution and Typological Divergence of County Population Shrinkage

#### 3.1.1. Spatial and Temporal Patterns of County Population Shrinkage

From 2000 to 2020, the trend of county population shrinkage in the North–South Transitional Zone of China was obvious. The number of counties with population shrinkage grew gradually, representing a change from “balanced evolution of expansion and shrinkage” to “dominated by shrinkage”, and the degree of shrinkage increased. According to the population shrinkage index, the 174 county-level administrative units in the study area could be divided into three types: significant shrinkage areas (*C_i_* ≤ −10%), general shrinkage areas (−10% < *C_i_* < 0), and non-shrinkage areas (*C_i_* > 0).

In terms of the study periods (Table 2), from 2000 to 2010, the number of shrinking and non-shrinking counties in the study area was basically the same, with 66 shrinking counties and 89 non-shrinking counties, accounting for 74.16% of the total number of county units; From 2010 to 2020, 73.6% of the counties in the study area experienced population shrinkage, mainly in the form of significant shrinkage, accounting for 51.56% of the total shrinking counties. In terms of administrative unit type, the five types of administrative units in the study area (municipal districts, county-level cities, counties, forest regions, and autonomous counties) all showed an evolutionary trend of population shrinkage, but the population shrinkage of different types of administrative units varied significantly. Due to the siphoning effect of urban activities, the population shrinkage of counties below the city level was the most prominent throughout the study period. By 2020, 86% of counties below the city level constituted shrinking areas; however, in the municipal districts with significant urban characteristics and functions, the population shrinkage was the most moderate, with 53.7% of the municipal districts being non-shrinking areas. The only forest district in the study area (Shennongjia) remained a population shrinkage area throughout the study period; the only autonomous county in the study (Beichuan Qiang nationality) evolved from a non-shrinking area to a shrinking area.

#### 3.1.2. Type Evolution of County Population Shrinkage

In order to confirm the population shrinkage of the counties at different administrative levels and explore the number and spatial pattern of the four types of counties, 174 counties in the North–South Transitional Zone of China were investigated (Table 3).

The results showed that most of the counties in the North–South Transitional Zone of China presented continuous shrinkage (79) or expansion followed by shrinkage (49), accounting for 45.4% and 28.2%, respectively. The number of counties presenting continuous expansion or shrinkage followed by expansion was small, with only 36 and 10, accounting for 20.7% and 5.7%, in these respective categories. In terms of the representatives of each shrinkage type, the geographic units of the continuous shrinkage type were mainly counties and county-level cities under the jurisdiction of cities, with 59 and 10 in each category, accounting for 55.1% and 41.7% of the counties and county-level cities under the jurisdiction of cities. The geographic units presenting expansion followed by shrinkage were dominated by counties below the city level, which indicated that the population shrinkage of counties below the city level was more common than that of other types of county units. The number of county units that shrank first and then expanded was very small, and these were represented by municipal districts, county-level cities, and counties under municipal jurisdiction. This showed that only a few types of county unit could reverse population shrinkage and become counties with population growth. This may be caused by the phenomenon of population shrinkage and the existence of path inertia. Population loss hinders regional development, and the slow development of regions continues to aggravate regional population losses. The continuously expanding county units were mainly municipal districts, such as Erqi District and Shangjie District. With rapid economic growth, complete infrastructure, and a strong population gathering capacity, the number of permanent residents in these areas showed a trend of continuous growth.

### 3.2. Spatial Differentiation of County Population Shrinkage

#### 3.2.1. Spatial Agglomeration and Gravity Shift of County Population Shrinkage

In order to explore the spatial distribution characteristics of population shrinkage in the counties of the North–South Transitional Zone of China, a spatial autocorrelation analysis of the population shrinkage index in the study area was conducted with counties as the unit. The results showed that the Moran’s I value of the population shrinkage index in the study area increased from 0.075 to 0.223, passing the *α* = 0.1 confidence test. This showed that the county population changes in the study area had significant positive spatial correlation, and the spatial agglomeration gradually increased.

In order to analyze the overall dynamic trend of the spatial evolution of population shrinkage in the study area, the standard deviation ellipse method was used to track the spatial gravity center and transfer path of the county population shrinkage and non-shrinkage areas, with the population shrinkage index as the weight (Figure 2). The results showed that from 2000 to 2020, the spatial gravity centers of the population shrinkage and non-shrinkage areas were consistently located in the eastern section of the Qinling Mountains. The spatial gravity center of the population-shrinkage area shifted from Hanbin District (32°55′ N, 108°37′ E) in Ankang City, Shaanxi Province to Shiquan County, Shaanxi Province (33°06′ N, 108°22′ E); that is, 36.89 km to the northwest. The spatial gravity center of the non-shrinkage area was transferred from Ningshan County, Ankang City, Shaanxi Province (33°30′ N, 108°38′ E) to Zhen’an County, Shangluo City, Shaanxi Province (33°33′ N, 109°13′ E); that is, 65.28 km to the northeast.

#### 3.2.2. “East–West” Corridor of County Population Shrinkage

The North–South Transitional Zone of China is an ecological corridor and an important social and economic channel connecting the east and west of China. As a comprehensive representation of regional development status, the spatial differentiation of population change directly reflected the corridor characteristics of the North–South Transitional Zone of China in the “east–west” direction. Using ArcGIS 10.3, the distance between the center points of the 174 counties after centroid abstraction and the datum line was calculated with the westernmost edge of the North–South Transitional Zone of China as the vertical datum line under the geodetic 2000 projection coordinates. In SPSS, we took the distance from the west datum line as the horizontal axis and the population shrinkage index as the vertical axis to draw a scatter plot and used the mean values of the horizontal and vertical coordinates as the benchmarks to draw the secondary coordinate axis, dividing the scatter plot into four quadrants (Figure 3).

The results showed moving from east to west across the North–South Transitional Zone of China, the population shrinkage index of each county presented a U-shaped evolution trend. The population shrinkage indices of the counties at the east and west ends of the transitional zone were significantly higher than those of the middle region. During the study period, about 51.8% of the counties whose population-shrinkage type was significant shrinkage were within 400~800 km from the west end of the study area. This showed that against the regional background of population shrinkage, the central region had a stronger population-loss tendency, which also conformed to the structural characteristics of “core periphery”.

#### 3.2.3. “North–South” Boundary of County Population Shrinkage

The Qinling–Huaihe Line, where the North–South Transitional Zone lies, is the dividing line between the northern and southern regions of China. Between the north and south sides of the line, there are obvious regional differences in natural geographical conditions, agricultural production, living customs, and regional socio-economic development. In order to explore the spatial differentiation of population shrinkage on both sides of the North–South Transitional Zone of China, which is dominated by the Qinling Mountains, the northern and southern foothills of the Qinling Mountains were divided by the Qinling Mountains watershed, and we took latitude as the horizontal axis, the population shrinkage index as the vertical axis, and the population shrinkage index and the average latitude of the Qinling Mountains watershed as the benchmarks to draw the secondary coordinate axis, dividing the scatter plot into four quadrants (Figure 4).

The results showed that there were significant differences between the counties with shrinking populations in the northern and southern foothills of the Qinling Mountains in the North–South Transitional Zone of China. From 2000 to 2020, 81.6% of the population-shrinkage counties were located to the south of the Qinling Mountains; furthermire, about 87% of the significant population-shrinkage counties were located to the south of the Qinling Mountains. This showed that the population-shrinkage counties in the North–South Transitional Zone of China were concentrated in the southern foothills of the Qinling Mountains in terms of spatial distribution during the study period, and the shrinkage degree was stronger in the south than in the north. The southern foothills were characterized by “concentrated spatial distribution and strong shrinkage degree”. This north–south differentiation was closely related to the topographic characteristics of the Qinling Mountains. The Qinling Mountains are characterized by a steep northern slope and a gentle southern slope. Therefore, there are a large number of mountainous counties in the southern foothills of the Qinling Mountains, such as Lueyang County and Liuba County in Shaanxi Province, whose population shrinkage indices were −28.54% and −23.69%, respectively. These counties suffered serious population shrinkage due to the impact of the terrain.

## 4. Analysis of the Factors Influencing County Population Shrinkage in the North–South Transitional Zone of China

### 4.1. Index selection

Regional population change mainly comprises a combination of natural population growth and population migration. Therefore, we considered these two factors in an integrated manner when exploring the causes of regional population shrinkage. Based on existing studies [43], we mainly considered physical geographic base, natural population growth, economic development, and infrastructure, and 21 influential indicators were selected to investigate the relationship between each factor and the county population shrinkage (Table 4). In order to consider the lag of the population shrinkage phenomenon, we used the data of the starting year for several socio-economic factors. Among the natural factors, the indicators of average elevation (*X*_1_), topographic relief (*X*_2_), and ecological environment (*X*_3_) were selected to represent the natural geographic basis of the study area, taking into full consideration the topographic characteristics of the North–South Transitional Zone of China, which is high in the west and low in the east. Among the human factors, population density (*X*_4_), natural growth rate (*X*_5_), aging rate (*X*_6_), and human capital (*X*_7_) were selected to characterize the regional population; regional GDP (*X*_8_), GDP per capita (*X*_9_), and disposable income per capita (*X*_10_) were selected to measure the level of regional economic development; non-agricultural industry output (*X*_11_) was selected to indicate the regional industrial structure; the share of non-agricultural employment (*X*_12_) was selected to characterize the regional employment opportunities; regional GDP growth (*X*_13_), GDP per capita growth (X_14_), and non-agricultural industry output value growth *(X*_15_) were selected to represent regional economic development speed; and urbanization level (*X*_16_), education level (*X*_17_), fiscal revenue (*X*_18_), road network density (*X*_19_), and medical facilities (*X*_20_) were used to reflect regional infrastructure and public service level.

### 4.2. Mechanisms of Population Loss Pattern Formation

#### 4.2.1. Comparison between Natural Factors and Human Factors

To clarify the influencing factors of population shrinkage in the North–South Transitional Zone of China, a geographic probe was used to detect the key driving factors and their explanatory power in the evolution of the spatial pattern of county population shrinkage from 2000 to 2020 (Figure 5). The results showed that the population shrinkage of counties in the North–South Transitional Zone of China was the result of the combined effect of natural and human factors, and there were differences in the degree of influence by natural and human factors. Among the natural factors, the mean altitude *(X*_1_), topographic relief (*X*_2_), and ecological environment (*X*_3_) passed the significance test with a q-mean of 0.125, while the human factors passed the significance test with a q-mean of 0.168, which was 34.4% higher than that of the natural factors, indicating that the influence of natural factors was generally weaker than that of human factors.

From 2000 to 2010, the q-mean of natural factors was 0.106, and the q-mean of human factors was 0.131; from 2010 to 2020, the q-mean of natural factors was 0.091, and the q-mean of human factors was 0.132. The results showed that the q-mean of natural factors decreased, while the q-mean of human factors increased, which indicated that there were significant regional differences in the natural geographic base and socio-economic level of the North–South Transitional Zone of China. The spatial pattern of population shrinkage in the counties was influenced by both natural and human factors; however, in general, the influence of natural factors was weaker than that of human factors, and the influence of natural factors decreased continuously, while that of human factors increased.

#### 4.2.2. Effect of Natural Factors

From 2000 to 2020, the natural factors of mean elevation (*X*_1_), topographic relief (*X*_2_), and ecological environment (*X*_3_) passed the significance test with q-values of 0.149, 0.146, and 0.080, respectively (Figure 4). This indicated that, across the whole population-shrinkage period, mean elevation (*X*_1_) and topographic relief (*X*_2_) played an important role in the regional population shrinkage. The ecological environment (*X*_3_) also played a role.

Natural geographic factors were the basis for the spatial pattern of population shrinkage in the counties in the North–South Transitional Zone of China, generally contributing to or constraining the evolution of county population shrinkage. According to the results of factor detection by period (Figure 6), among the natural factors, average elevation (*X*_1_) and topographic relief (*X*_2_) were the most important natural influences, with q-means of 0.115 and 0.107, respectively, and the q-mean of ecological environment (*X*_3_) was 0.074. This indicated that topographic conditions had a prominent effect on the population shrinkage pattern in the study area, and that the ecological environment also had a significant effect on county population shrinkage. Among the natural factors, topographic relief (*X*_2_) was an enhanced factor, with its q-mean increasing from 0.101 to 0.114; mean elevation (*X*_1_) and ecological environment (*X*_3_) were weakened factors, with their q-mean values decreasing from 0.137 and 0.081 to 0.093 and 0.066, respectively. The topographic conditions of the transition zone, which is high in the west and low in the east and dominated by mountainous terrain, continuously exacerbated the regional population shrinkage.

In order to accurately portray the influence of topographic conditions on population shrinkage in the North–South Transitional Zone of China, two natural-factor indicators with similar q-values, namely, average elevation (*X*_1_) and topographic relief (*X*_2_), were selected, and population shrinkage contour maps of the counties were drawn for the whole study period with average elevation (*X*_1_) as the vertical axis and topographic relief (*X*_2_) as the horizontal axis (Figure 7). The results showed that from 2000 to 2020, the average elevation of counties with significant population shrinkage was concentrated between 1000 and 1500 m, and their topographic relief was above 1500 m, which indicated that topographic factors played an important role in the overall population shrinkage in the North–South Transitional Zone of China. For example, the population shrinkage indices of the Qingchuan and Pingwu counties in Sichuan province were −38.29% and −32.72%, respectively; their average elevation was above 1200 m; and their topographic undulations are evenwere above 3200 m and 4100 m, respectively.

The contour map of population shrinkage by time period showed that the natural factors in the North–South Transitional Zone of China had a strong endogenous rooting effect on the formation of the spatial pattern of population shrinkage (Figure 8), and topographic relief (*X*_2_) and average altitude (*X*_1_) had an important influence on the population shrinkage of each county. From 2000 to 2010, the average altitude of the shrinking counties was concentrated between 500 and 1500 m, and the topographic relief was between 1500 and 3500 m, while the counties with an average altitude above 2500 m were in a state of population expansion. This was because the counties with a higher altitude are mainly located in the western part of the North–South Transitional Zone of China, which is home to ethnic minorities, whose traditions cause these counties to be sparsely populated. For example, the total fertility rate of the Aba Prefecture in Sichuan Province and the Linxia Prefecture in Gansu Province were 1.57 and 1.72, respectively, at the time of the Sixth Five-Year Plan, more than 30% higher than the total fertility rate of their provinces. From 2010 to 2020, counties with population shrinkage expanded rapidly in areas with a high altitude and considerable topographic relief, and population shrinkage occurred in all counties with topographic relief over 3000 m. This indicated that the driving influence of topographic relief on regional population shrinkage was significantly enhanced, and the population shrinkage in counties constrained by both high altitude and high topographic relief (most of which were persistently shrinking counties) was more serious than that in counties constrained by a single topographic factor. For example, the Pingwu and Wenchuan counties in Sichuan province were both persistently shrinking counties, with population shrinkage indices of −32.72% and −25.88%, respectively, during the study period. The average elevation of these two counties was above 2000 m, and their topographic relief was above 4100 m and 4500 m, respectively. From the comparison of the two study periods, the average altitude and topographic relief of the counties with significant population shrinkage in the North–South Transitional Zone of China increased over time, and the counties with a high altitude and high topographic relief gradually became the hardest-hit areas for population shrinkage.

#### 4.2.3. Dominant Role of Human Factors

Human factors were the basic driving force behind the spatial pattern of population shrinkage in the counties in the North–South Transitional Zone of China (Figure 5 and Figure 6). Across the whole study period, from 2000 to 2020, among the human factors, education level (*X*_17_), urbanization level (*X*_16_), and road network density (*X*_19_) were the main influencing factors for the spatial pattern of county population shrinkage in the North–South Transitional Zone of China, with q-values of 0.301, 0.273, and 0.245, respectively. Human capital (*X*_7_), the proportion of non-farm employment (*X*_12_), and population density (*X*_4_) were the secondary influences on the evolution of the county population shrinkage pattern, with q-values of 0.211, 0.210, and 0.209, respectively. Finally, non-agricultural industry output growth (*X*_15_), GDP per capita (*X*_9_), and GDP per capita growth (*X*_13_) were general influences on the spatial pattern of county population shrinkage, with q-values of 0.132, 0.129, and 0.109, respectively. In the North–South Transitional Zone of China, where the economic development is poor and poverty is prominent, a low education level, lagging urbanization, and poor transportation conditions were the key influencing factors behind the spatial pattern of regional population shrinkage. Talent resources, non-agricultural industry development, and population density also influenced regional population shrinkage, while talent resources could provide support for regional industrial upgrading and non-agricultural industry development, generating more jobs to absorb surplus labor.

To further explore the driving mechanism of the key factors on regional population shrinkage, the four influencing factors with the largest factor detection values, namely education level (*X*_17_), road network density (*X*_19_), urbanization level (*X*_16_), and non-farm employment share (*X*_11_), were selected and further analyzed with the help of scatter plots (Figure 9).

The average years of schooling can reflect the education level of a region, and most of the counties in the study area presenting population shrinkage had lower average years of schooling. Of the 79 counties with below-average schooling, 45 experienced population shrinkage from 2000 to 2010, accounting for 56.9%; 74 of the 86 counties with below-average schooling experienced population shrinkage from 2010 to 2020, accounting for 88.1%. This indicated that education level had a positive effect on regional population shrinkage, and a higher education level could provide a suitably high-quality labor force to accelerate regional industrial development and economic transformation, thus promoting regional technological development and social progress and curbing regional population shrinkage.

The road network density reflects the level of regional transportation facilities, and most of the counties with poor road traffic conditions experienced population shrinkage during the study period. Of the 133 counties with a lower-than-average road network density, 79 experienced population shrinkage from 2000 to 2010, accounting for 59.4%; 111 of the 131 counties with a lower-than-average road network density experienced population shrinkage from 2010 to 2020, accounting for 84.7%. This indicated that counties with weak infrastructure and prominent transportation constraints in the Qinba Mountains were the hardest-hit by population shrinkage. In the Qinba Mountains, transportation is a solid foundation for economic and social development, and the improvement of transportation facilities is of great significance for alleviating population shrinkage in the region.

The level of urbanization is an important indicator of regional modernization, reflecting the domestic demand potential and development momentum of a region, and the process of urbanization construction is also the process of releasing momentum for regional economic development. From 2010 to 2020, 93 out of 112 counties with lower-than-average urbanization rates experienced population shrinkage, accounting for 80%. This indicated that urbanization construction had a positive effect on regional population shrinkage, and new urbanization construction could effectively curb regional population shrinkage.

The proportion of non-farm employment can be used to measure the industrial structure of a region, with the secondary and tertiary industries able to provide more employment opportunities and a higher economic income, accommodating the surplus labor from the primary industry in the region. Of the 120 counties with a lower-than-average proportion of non-farm employment, 71 experienced population shrinkage from 2000 to 2010, accounting for 59.2%; from 2010 to 2020, 109 counties with a lower-than-average proportion of non-farm employment experienced population shrinkage. From 2010 to 2020, 90 of the 109 counties with below-average non-farm employment were in a state of population shrinkage, accounting for 82.6%. This indicated that there were insufficient non-farm employment opportunities in the shrinking counties to satisfy the non-farm employment demand in the region, which caused segments of the population who had difficulty obtaining local jobs to flow to more economically and socially developed areas with more employment opportunities, thus exacerbating regional population shrinkage.

#### 4.2.4. Human–Nature Synergy

An interaction detector was used to clarify the synergistic relationships among the influencing factors of population shrinkage in the North–South Transitional Zone of China. The results showed that the synergy among human factors was the key driving force behind the county-level shrinkage pattern in the North–South Transitional Zone of China, and the synergy between natural and human factors had an important influencing role (Table 5).

From 2000 to 2020, the interaction between natural factors and human, economic, and other factors in the North–South Transitional Zone of China was stronger than the interactions within these factor groups, showing both non-linear enhancement and two-factor enhancement. The synergy between population density (*X*_4_) and urbanization level (*X*_16_), ecological environment (*X*_3_) and education level (*X*_17_), and human capital (*X*_9_) and road network density (X_19_) had prominent effects on regional population shrinkage, with interaction detection values of 0.464, 0.454, and 0.446, respectively. The synergy between non-farm industry output growth (*X*_15_) and urbanization level (*X*_16_), average altitude (*X*_1_) and human capital (*X*_9_), and natural population growth rate (*X*_5_) and education level (*X*_17_) had strong effects on regional population shrinkage, with interaction detection values of 0.427, 0.425, and 0.422, respectively. The synergy between population density (*X*_6_) and education level (*X*_17_), GDP per capita (*X*_9_) and education level (*X*_17_), and the share of non-farm employment (X_2_) and urbanization level (*X*_16_) had important effects on regional population shrinkage, with interaction detection values of 0.408, 0.406, and 0.401, respectively. From a broad temporal perspective, the vast spatial area and complex topography of the North–South Transitional Zone of China have formed unique physical geographic features and socio-economic structures. The synergistic interactions between human factors were the key drivers behind the evolution of the population-shrinkage spatial pattern, among which urbanization level (*X*_16_) and education level (*X*_17_) interacted widely with other factors. This indicated that the synergy between urbanization construction and education level and other factors in the North–South Transitional Zone of China had the most crucial influence on regional population shrinkage. The reason for this was that urbanization construction can attract surplus rural populations into cities and towns by increasing the employment opportunities and industrial development, and education level improvement can provide a large pool of high-quality talent for regional industrial upgrading.

The results of the influencing factor interaction detection by time period showed that the interaction and synergy of natural and human factors were the key drivers of the county population-shrinkage spatial pattern in the North–South Transitional Zone of China (Table 6). During the study period, the interaction between natural factors and human and economic factors in the North–South Transitional Zone of China was stronger than the interaction within these factor groups, showing both non-linear and bi-factor enhancement. From 2000 to 2010, the synergy between ecological environment (*X*_3_) and education level (*X*_17_), mean altitude (*X*_1_) and population density (*X*_4_), non-agricultural industry output growth (*X*_15_) and education level (*X*_17_), human capital (*X*_7_) and road network density (*X*_19_), average altitude (*X*_1_) and non-agricultural output growth (*X*_15_), and human capital (*X*_7_) and GDP per capita (*X*_9_) had prominent effects on regional population shrinkage, with interaction detection values of 0.475, 0.473, and 0.457, respectively. The synergy between human capital (X_7_) and road network density (*X*_19_), average altitude (*X*_1_) and non-agricultural output growth (X_15_), and human capital (*X*_7_) and GDP per capita (*X*_9_) had strong effects on regional population shrinkage, with interaction detection values of 0.446, 0.428, and 0.400, respectively. The synergy between the natural population growth rate (*X*_5_) and population density (*X*_4_) and between human capital (*X*_7_) and education level (*X*_17_) had a significant effect on regional population shrinkage, with interaction detection values of 0.390, 0.386, and 0.385, respectively. From 2010 to 2020, the synergy between human capital (*X*_7_) and GDP per capita (*X*_9_), human capital (*X*_7_) and education level (*X*_17_), and non-farm employment share (*X*_12_) and road network density (*X*_19_) had prominent effects on population shrinkage, with interaction detection values of 0.363, 0.358, and 0.331, respectively. The synergy between regional GDP (*X*_8_) and urbanization level (*X*_16_), aging rate (*X*_6_) and education level (*X*_17_), and population density (*X*_4_) and urbanization level (*X*_16_) played a significant role, with interaction detection values of 0.329, 0.32, and 0.319, respectively. The synergy between aging rate (*X*_6_) and urbanization level (*X*_16_), GDP per capita (*X*_9_) and education level (*X*_17_), and regional GDP (*X*_8_) and education level (*X*_17_) also played a significant role.

The vast spatial area and complex topography in the North–South Transitional Zone of China have created unique physical geographic features and socio-economic structures. From 2000 to 2010, the synergy between natural factors and human factors played a fundamental role, and the interaction between education level and other human factors also had a strong influence. From 2010 to 2020, the synergistic effect of natural factors and human factors weakened, and the synergistic interactions between human factors were the key drivers of the evolution of the population-shrinkage spatial pattern, among which interactions between education level, human capital, and other factors were common. This indicated that the interaction between the natural environment of the North–South Transitional Zone of China, which has a high elevation in the west and a low elevation in the east, and the education level and industrial structure was the driving force behind the regional population shrinkage, and the interactions of education level, human capital, and road network density with other human factors played an important role in the formation and evolution of the regional population-shrinkage pattern.

## 5. Discussion

### 5.1. Diverse and Complex Factors Influencing Population Shrinkage in Less-Developed Mountainous Areas

As a developing country, with the rapid advancement of urbanization, population shrinkage in China at the county level and below has become a widespread phenomenon, appearing most commonly in less-developed mountainous areas [44,45]. The factors influencing population shrinkage are complex and diverse, with the natural geographic environment playing a fundamental role [46] and having a profound endogenous rooting effect on regional resource development and transportation improvement. However, most existing studies attribute the population shrinkage phenomenon to socio-economic factors and ignore the important role of natural factors such as altitude, topographic relief, and the ecological environment. To address the problem of previous studies focusing too much on the phenomenon of population shrinkage with insufficient explanation of its influencing factors, we considered the North–South Transitional Zone of China as a case study; adopted spatial-mapping and geographic exploration methods; constructed a systematic research framework for the analysis of the spatial and temporal evolution of population shrinkage and its influencing factors in less-developed mountainous areas; and explored the influence of the natural geographic background, socio-economic development, and human–land synergy on regional population shrinkage. Our research results not only enrich the theoretical understanding of regional population shrinkage, but also provide a basis for the scientific explanation of the development trends of less-developed mountainous areas. Our findings could also provide inspiration for the transformation and development of similar areas.

Population evolution is the combined result of socio-economic, ecological, and institutional/policy factors, but the degree of population shrinkage and its major influencing factors vary between different countries and regions [47,48]. The population shrinkage in parts of Europe and the United States has been brought about by the economic transformation from manufacturing to service industries and the resulting transfer of labor and industrial capital [11]. In addition, policy migration, natural and human-made disasters, low fertility rates, and the breakdown of political systems can also lead to population shrinkage [29,44]. However, this study found that education level and road network density had the most significant effect on population shrinkage in the North–South Transitional Zone of China, contrasting with other studies that identified the socio-economic development level as having the strongest effect on population shrinkage. This was probably because of the poor education level and weak transportation infrastructure in the North–South Transitional Zone of China, resulting in a lack of high-quality labor for regional industrial development and upgrading, and suboptimal transportation conditions, which induced serious population shrinkage. Therefore, it is of great practical significance to continue to increase regional investment in science and education and continuously improve the regional transportation network in the study area, so that the North–South Transition Zone of China can give full play to its locational advantages, being at the east–west and north–south intersections.

### 5.2. Population Shrinkage Is A New Trend in the Demographic Evolution of Less-Developed Areas

Imbalances in interregional development can lead to imbalances in interregional population movements [47], and despite the implementation of several strategies by the central government to promote coordinated regional development, imbalances in policy support still exist. Studies have found, for example, an obvious trend of population shrinkage in less-developed regions such as Ireland and Estonia, which may become an important trend in regional demographic evolution for a long time to come, seriously threatening sustainable regional economic and social development [43]. It is difficult for less-developed regions to maintain their advantages in competition with developed regions, and less-developed regions will probably remain the key areas of population shrinkage.

The population shrinkage of counties in the North–South Transitional Zone of China was mostly represented by continuous shrinkage, indicating that the population shrinkage in these counties is characterized by cyclical accumulation [49], i.e., once population shrinkage occurs, it is difficult to reverse the trend in the vast majority of counties, a finding that concurred with the results of existing studies [50]. In addition, the current population fertility rate in China continues to decline, and projections for the future population fertility situation in the North–South Transitional Zone of China are not optimistic, which will continue to worsen the problem of regional population shrinkage in the long run. In light of our research findings, we propose the following policy recommendations to address the regional population shrinkage problem: First, local governments need to pay attention to the regional population shrinkage problem and its impact on regional economic and social development. On the one hand, increasing investment in education and transportation is the key for dealing with the regional population shrinkage problem; on the other hand, governments should take advantage of mountainous areas and make effective use of local ecological and tourism resources. Secondly, local governments need to take targeted measures to curb the continuous outflow of regional populations. The central and western regions, which are economically disadvantaged, need to improve their economic development, enhance their employment capacity, and increase the attractiveness of urbanized areas to the local population; the eastern regions need to promote further economic development through industrial transformation and take measures to encourage childbirth.

### 5.3. Limitations and Prospects

Due to limitations of space, although this study analyzed the influencing factors of population shrinkage in less-developed mountainous areas, we did not explore in depth the driving mechanisms and interaction mechanisms between socio-economic and natural environment factors across different development stages, which is of great significance in the study of regional population shrinkage; thus, it is necessary to conduct further in-depth research to strengthen the theoretical understanding of regional population shrinkage. Due to the limited microdata acquisition, this paper focused on the effects of natural factors, human factors, and human–nature interactions on population shrinkage, while we did not cover social factors such as marriage and fertility attitudes; policy factors such as family planning and ecological migration; or the effects of changes in specific groups such as the mobile population, highly educated people, the elderly population, and the ethnic minority population on population shrinkage. Additional studies should be conducted in the future according to the level of detail of the population census data. In addition, the population shrinkage characteristics and their influencing factors display scale effects, and we only analyzed and discussed these elements based on the county-level scale; in the future, census data from townships and streets could be used to verify the variability of the findings at different scales.

## 6. Conclusions

Based on standard deviation ellipse and other methods, we analyzed the spatio-temporal characteristics of the population-shrinkage regions and counties in the North–South Transitional Zone of China from 2000 to 2020 and, on this basis, used geographic indicators to detect the driving factors of county population shrinkage in this area across different periods. The main conclusions were as follows:

(1) From 2000 to 2020, the number of population-shrinkage counties in the North–South Transitional Zone of China increased, gradually evolving from “balanced evolution of expansion and shrinkage” to “dominated by shrinkage”, and the degree of shrinkage also increased. In terms of administrative unit types, counties with population shrinkage were mainly municipal counties. In terms of shrinkage type, county units mainly displayed continuous shrinkage or expansion followed by shrinkage; municipal counties were the main representatives of both types of shrinkage, and they were more likely to present population shrinkage phenomena than other types of county units.

(2) In terms of spatial differentiation, the population shrinkage of counties in the North–South Transitional Zone of China demonstrated significant spatial autocorrelation and tended to increase over time, with an obvious pattern of contiguous shrinkage within county units. The center of gravity of shrinkage and the shrinkage agglomeration areas both showed an evolutionary trend from east to west. In the “east–west” corridor, the population shrinkage index of each county and region showed a U-shaped trend, and the population shrinkage index of the counties at the east and west ends of the North–South Transitional Zone of China was significantly higher than that of the middle region; therefore, in the context of overall population shrinkage, the central region had a stronger tendency to decrease in population size, which was also in line with the “core–periphery” structural characteristics. In terms of “south–north” divergence, due to the topographic characteristics of the Qinling Mountains, the population shrinkage in the study area was concentrated in the southern part of the Qinling Mountains, and the degree of population shrinkage here was stronger than that in the northern part of the Qinling Mountains.

(3) Natural geographic factors had a significant endogenous rooting effect on population shrinkage in counties in the North–South Transitional Zone of China, while human factors had a strong influencing effect on the evolution of regional population shrinkage patterns, with significant differences in the effects of different factors. Overall, the role of natural factors was weaker than that of human factors. Among the natural factors, the constraint of terrain undulation degree increased, and districts and counties with a high altitude and high terrain undulation gradually became the hardest-hit by population shrinkage. Among the human factors, education level and urbanization level played an important role in regional population shrinkage, while prominent transportation constraints and fewer non-farm employment opportunities also aggravated regional population shrinkage.

(4) The formation and evolution of the spatial pattern of population shrinkage in counties were subject to the synergistic effects of natural and human factors, and the interaction between natural factors and human economic factors was stronger than the interaction within factors, and showed both nonlinear enhancement and bi-factor enhancement. The interactions between the natural background of the Transitional Zone of China, which is high in the west and low in the east, and the backward education level and industrial structure are the key factors inducing the regional population contraction, while the interactions between the human factors, such as education level, human capital and road network density, and other human factors play an important driving role in the formation and evolution of the regional population shrinkage pattern.

## Figures and Tables

**Figure 1 ijerph-19-15801-f001:**
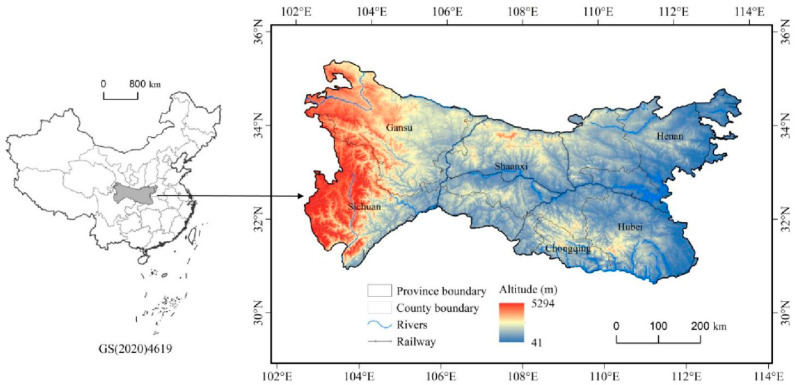
Location of the North–South Transitional Zone of China.

**Figure 2 ijerph-19-15801-f002:**
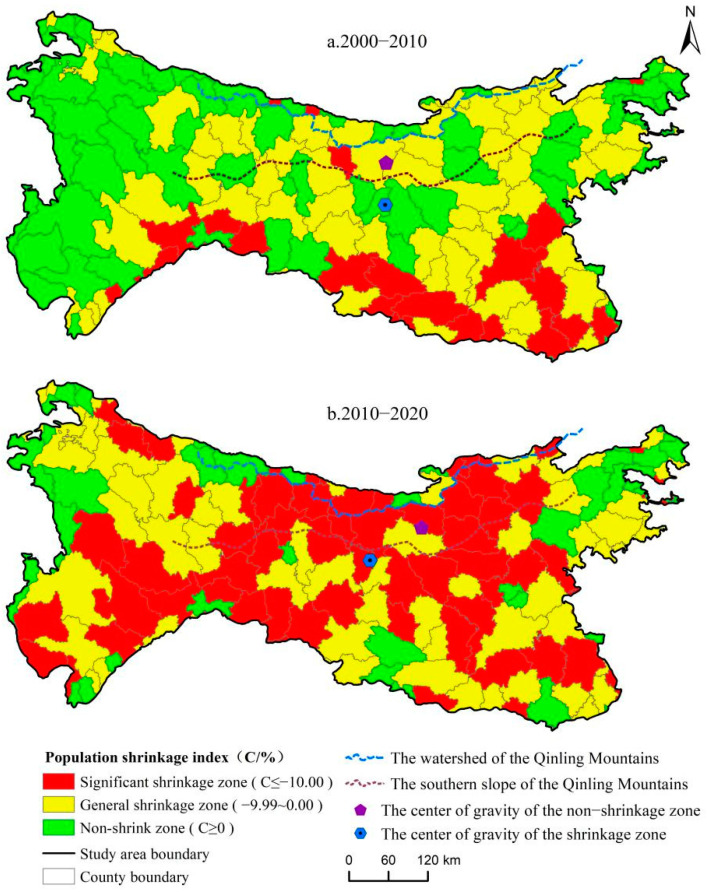
County population shrinkage patterns in North–South Transitional Zone of China (**a**,**b**) (2000–2020).

**Figure 3 ijerph-19-15801-f003:**
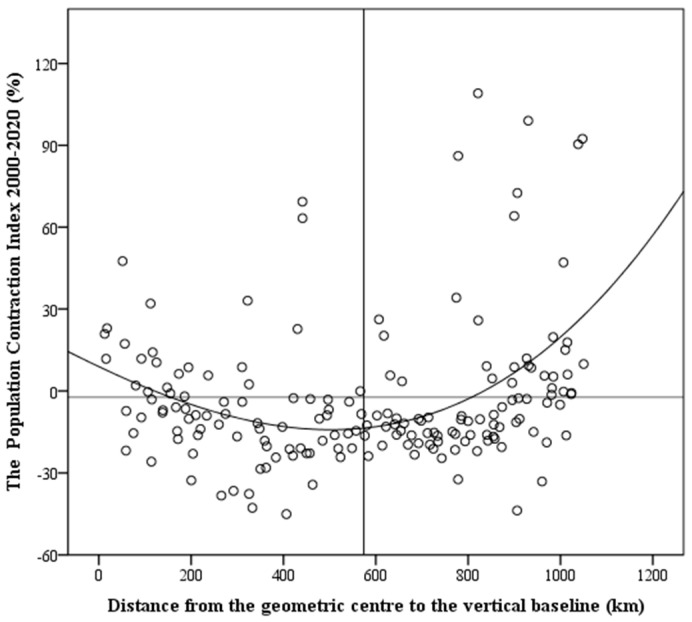
The “east–west” changes in population shrinkage index in the North−South Transitional Zone of China (2000−2020).

**Figure 4 ijerph-19-15801-f004:**
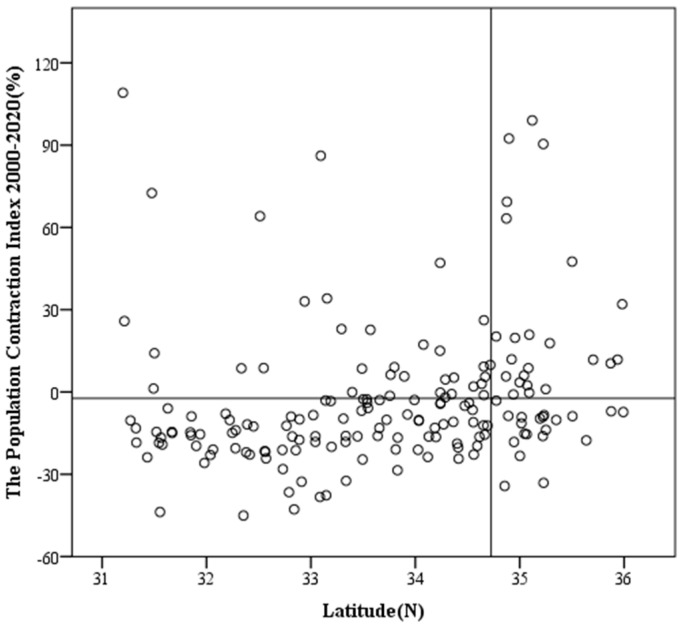
The “north−south” changes in the population shrinkage index in the North−South Transitional Zone of China (2000−2020).

**Figure 5 ijerph-19-15801-f005:**
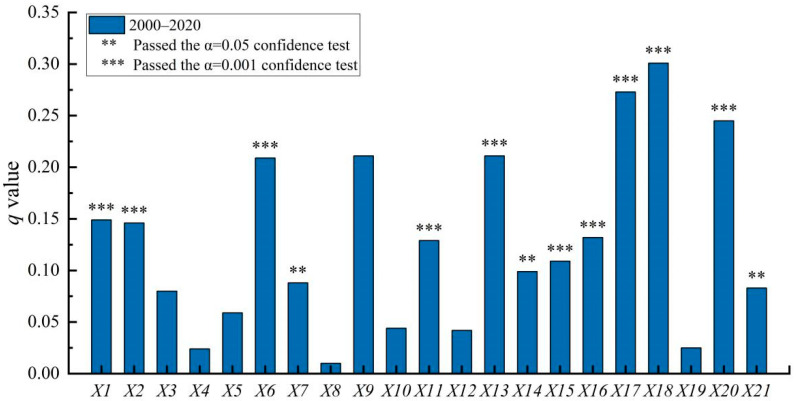
Contribution rate of population-shrinkage influencing factors in the North–South Transitional Zone of China (2000–2020).

**Figure 6 ijerph-19-15801-f006:**
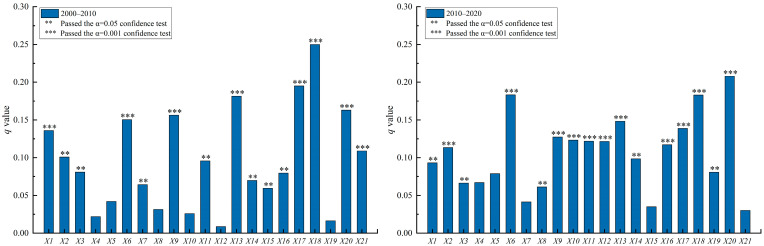
Contribution rate of influencing factors of population shrinkage in the North–South Transitional Zone of China by time period.

**Figure 7 ijerph-19-15801-f007:**
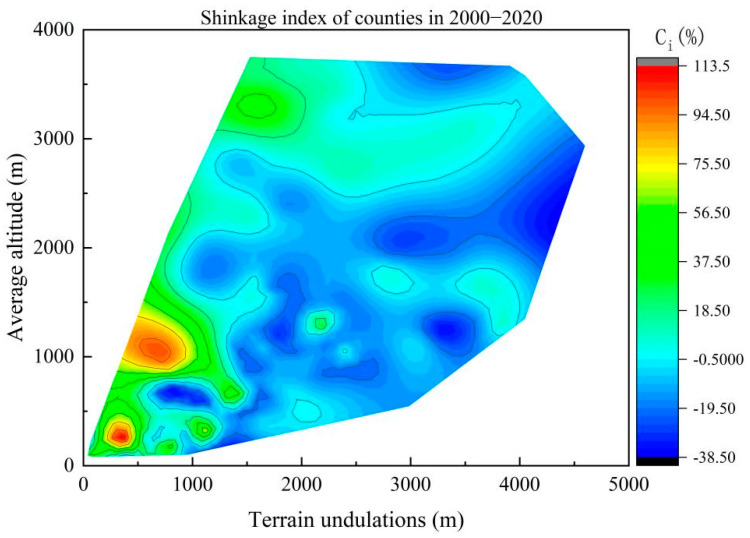
Contour map of county population shrinkage for the whole study period in the North−South Transitional Zone of China.

**Figure 8 ijerph-19-15801-f008:**
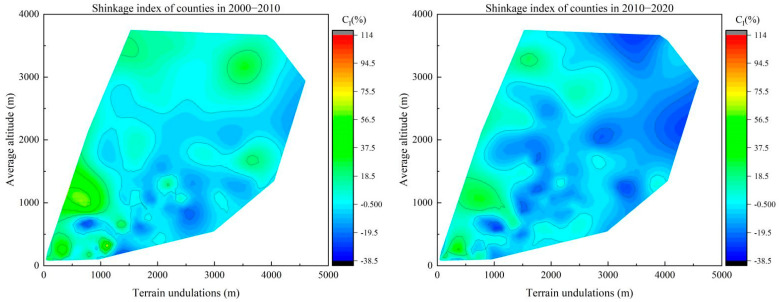
Contour map of county population shrinkage in the North−South Transitional Zone of China by time period.

**Figure 9 ijerph-19-15801-f009:**
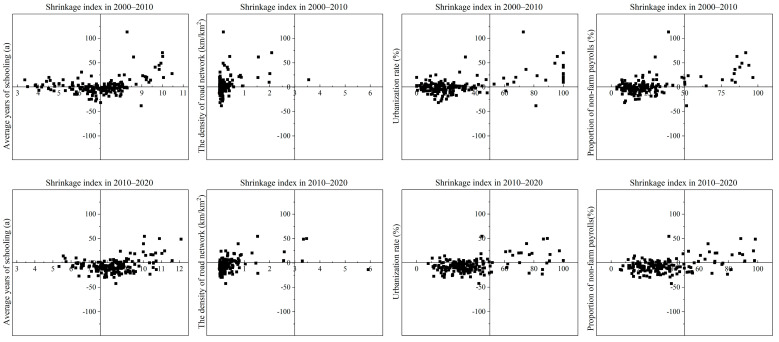
Scatter diagram of four human influencing factors in the North–South Transitional Zone of China.

**Table 1 ijerph-19-15801-t001:** Types of interaction between two factors.

Judgment Basis	Interaction Type
*q*(*X*_1_) ∩ *q*(*X*_2_) * < Min (*q*(*X*_1_), *q*(*X*_2_))	Non-linear attenuation
Min (*q*(*X*_1_), *q*(*X*_2_)) < *q*(*X*_1_) ∩ *q*(*X*_2_) < Max (*q*(*X*_1_*)*, *q*(*X*_2_))	One-factor non-linear enhancement
*q*(*X*_1_) ∩ *q*(*X*_2_) > Max (*q*(*X*_1_), *q*(*X*_2_))	Bi-factor enhancement
*q*(*X*_1_) ∩ *q*(*X*_2_) = *q*(*X*_1_) + *q*(*X*_2_)	Independent
*q*(*X*_1_) ∩ *q*(*X*_2_) > *q*(*X*_1_) + *q*(*X*_2_)	Non-linear enhancement

* In the table, *q*(*X*_1_) ∩ *q*(*X*_2_) is the *q* value of interaction between two geographical factors, and *q*(*X*_1_) + *q*(*X*_2_) is the sum of *q* values of two independent geographical factors.

**Table 2 ijerph-19-15801-t002:** The number of shrinking county units in the North–South Transitional Zone of China in different periods.

Period		Types of County with Population Shrinkage					Total
		Municipal District	County-Level City	County	Forest District	Autonomous County	
2000–2010	Quantity	13	12	63	1	0	89
	Proportion (%)	32	50	59	100	0	51
2010–2020	Quantity	19	15	92	1	1	128
	Proportion (%)	46	64	86	100	100	74

**Table 3 ijerph-19-15801-t003:** Typological characteristics of county population shrinkage.

Type	Number of County Units					Total	Proportion
	Municipal District	County-Level City	County	Forest District	Autonomous County		
Continuous shrinkage	9	10	59	1	0	79	45.4%
Shrinkage followed by expansion	4	2	4	0	0	10	5.7%
Expansion followed by shrinkage	10	5	33	0	1	49	28.2%
Continuous expansion	18	7	11	0	0	36	20.7%

**Table 4 ijerph-19-15801-t004:** Selection and description of influencing factors of county population shrinkage.

Influencing Factors	Indicators	Variable	Description
Natural Foundations	Average altitude	*X* _1_	Calculated from the DEM
	Terrain undulation	*X* _2_	Calculated from the DEM
	Ecological environment	*X* _3_	Expressed by the NDVI index
Population Characteristics	Population density	*X* _4_	Population density in the starting year
	Natural growth rate	*X* _5_	Natural population growth rate of the region in the starting year
	Ageing rate	*X* _6_	Proportion of population aged 65 or above in the starting year
	Human capital	*X* _7_	Number of residents with a college education or above in the starting year
Economic Development	Regional GDP	*X* _8_	Regional GDP in the starting year
	GDP per capita	*X* _9_	GDP per capita in the starting year
	Disposable income per capita	*X* _10_	Disposable income per capita in the starting year
	Non-agricultural industry output	*X* _11_	Non-agricultural output in the starting year
	Non-agricultural employment share	*X* _12_	Share of population employed in non-agricultural industries in the starting year
	Regional GDP growth	*X* _13_	Increase in regional GDP at the end of the period compared to the beginning of the period
	GDP per capita growth	*X* _14_	Increase in GDP per capita at the end of the period compared to the beginning of the period
	Growth in non-agricultural output	*X* _15_	Increase in non-agricultural industry output value at the end of the period compared to the beginning of the period
Infrastructure	Urbanization level	*X* _16_	Share of urban population in the starting year
	Education level	*X* _17_	Average years of schooling in the starting year
	Fiscal revenue	*X* _18_	Fiscal revenue in the starting year
	Road network density	*X* _19_	Ratio of road network length to administrative area in counties
	Medical facilities	*X* _20_	Number of hospital beds per 10,000 people in the starting year

**Table 5 ijerph-19-15801-t005:** Interaction detection results for factors influencing population shrinkage in the North–South Transitional Zone of China.

Period	A ∩ B	*q* (A ∩ B)	Type	A ∩ B	*q* (A ∩ B)	Type	A ∩ B	*q* (A ∩ B)	Type
2000–2020	*X*_1_ ∩ *X*_4_	0.364	BE	*X*_1_ ∩ *X*_15_	0.428	NE	*X*_1_ ∩ *X*_17_	0.339	BE
	*X*_1_ ∩ *X*_7_	0.325	BE	*X*_1_ ∩ *X*_13_	0.426	NE	*X*_1_ ∩ *X*_18_	0.316	BE
	*X*_1_ ∩ *X*_6_	0.237	NE	*X*_1_ ∩ *X*_12_	0.308	BE	*X*_1_ ∩ *X*_20_	0.306	BE
	*X*_2_ ∩ *X*_7_	0.312	BE	*X*_2_ ∩ *X*_12_	0.281	BE	*X*_2_ ∩ *X*_18_	0.383	BE
	*X*_2_ ∩ *X*_6_	0.258	BE	*X*_2_ ∩ *X*_15_	0.296	NE	*X*_2_ ∩ *X*_17_	0.363	BE
	*X*_2_ ∩ *X*_4_	0.255	NE	*X*_2_ ∩ *X*_11_	0.291	NE	*X*_2_ ∩ *X*_20_	0.351	BE
	*X*_3_ ∩ *X*_9_	0.331	NE	*X*_3_ ∩ *X*_13_	0.344	NE	*X*_3_ ∩ *X*_18_	0.454	NE
	*X*_3_ ∩ *X*_6_	0.263	BE	*X*_3_ ∩ *X*_16_	0.291	NE	*X*_3_ ∩ *X*_17_	0.387	NE
	*X*_3_ ∩ *X*_7_	0.208	NE	*X*_3_ ∩ *X*_15_	0.288	NE	*X*_3_ ∩ *X*_20_	0.338	NE
	*X*_9_ ∩ *X*_11_	0.364	NE	*X*_9_ ∩ *X*_18_	0.385	BE	*X*_13_ ∩ *X*_17_	0.401	BE
	*X*_9_ ∩ *X*_15_	0.356	NE	*X*_9_ ∩ *X*_17_	0.325	BE	*X*_13_ ∩ *X*_20_	0.396	BE
	*X*_9_ ∩ *X*_12_	0.333	NE	*X*_9_ ∩ *X*_20_	0.446	BE	*X*_13_ ∩ *X*_18_	0.393	BE
	*X*_6_ ∩ *X*_13_	0.364	BE	*X*_6_ ∩ *X*_17_	0.464	BE	*X*_16_ ∩ *X*_17_	0.427	NE
	*X*_6_ ∩ *X*_15_	0.340	NE	*X*_6_ ∩ *X*_18_	0.408	BE	*X*_16_ ∩ *X*_18_	0.360	E
	*X*_6_ ∩ *X*_10_	0.316	NE	*X*_6_ ∩ *X*_21_	0.383	NE	*X*_16_ ∩ *X*_20_	0.317	E
	*X*_7_ ∩ *X*_12_	0.331	NE	*X*_7_ ∩ *X*_18_	0.422	NE	*X*_11_ ∩ *X*_18_	0.406	E
	*X*_7_ ∩ *X*_15_	0.313	NE	*X*_7_ ∩ *X*_17_	0.387	NE	*X*_11_ ∩ *X*_17_	0.381	E
	*X*_7_ ∩ *X*_11_	0.285	NE	*X*_7_ ∩ *X*_20_	0.334	E	*X*_11_ ∩ *X*_20_	0.332	E

NE indicates non-linear enhancement, and BE indicates bi-factor enhancement.

**Table 6 ijerph-19-15801-t006:** Interaction detection results for factors influencing population shrinkage by time period in the North–South Transitional Zone of China.

Period	A ∩ B	*q* (A ∩ B)	Type	A ∩ B	*q* (A ∩ B)	Type	A ∩ B	*q* (A ∩ B)	Type
2000–2010	*X*_1_ ∩ *X*_9_	0.341	NE	*X*_1_ ∩ *X*_13_	0.308	BE	*X*_1_ ∩ *X*_18_	0.339	BE
	*X*_1_ ∩ *X*_6_	0.473	NE	*X*_1_ ∩ *X*_11_	0.202	BE	*X*_1_ ∩ *X*_17_	0.274	BE
	*X*_1_ ∩ *X*_7_	0.280	NE	*X*_1_ ∩ *X*_16_	0.428	NE	*X*_1_ ∩ *X*_20_	0.256	BE
	*X*_2_ ∩ *X*_9_	0.312	NE	*X*_2_ ∩ *X*_13_	0.281	BE	*X*_2_ ∩ *X*_18_	0.309	BE
	*X*_2_ ∩ *X*_6_	0.255	BE	*X*_2_ ∩ *X*_11_	0.247	NE	*X*_2_ ∩ *X*_17_	0.261	BE
	*X*_2_ ∩ *X*_7_	0.223	NE	*X*_2_ ∩ *X*_16_	0.244	NE	*X*_2_ ∩ *X*_20_	0.294	NE
	*X*_3_ ∩ *X*_9_	0.311	NE	*X*_3_ ∩ *X*_13_	0.355	NE	*X*_3_ ∩ *X*_18_	0.475	NE
	*X*_3_ ∩ *X*_6_	0.246	NE	*X*_3_ ∩ *X*_11_	0.245	NE	*X*_3_ ∩ *X*_17_	0.371	NE
	*X*_3_ ∩ *X*_7_	0.223	NE	*X*_3_ ∩ *X*_16_	0.200	NE	*X*_3_ ∩ *X*_20_	0.280	NE
	*X*_9_ ∩ *X*_13_	0.350	NE	*X*_7_ ∩ *X*_18_	0.385	BE	*X*_13_ ∩ *X*_18_	0.338	BE
	*X*_9_ ∩ *X*_11_	0.400	NE	*X*_7_ ∩ *X*_17_	0.325	BE	*X*_13_ ∩ *X*_17_	0.330	BE
	*X*_9_ ∩ *X*_16_	0.219	BE	*X*_7_ ∩ *X*_20_	0.446	NE	*X*_13_ ∩ *X*_20_	0.352	BE
	*X*_6_ ∩ *X*_13_	0.369	NE	*X*_6_ ∩ *X*_18_	0.386	BE	*X*_11_ ∩ *X*_18_	0.378	NE
	*X*_6_ ∩ *X*_11_	0.292	NE	*X*_6_ ∩ *X*_17_	0.371	NE	*X*_11_ ∩ *X*_17_	0.308	NE
	*X*_6_ ∩ *X*_16_	0.191	BE	*X*_6_ ∩ *X*_20_	0.349	NE	*X*_11_ ∩ *X*_20_	0.243	BE
	*X*_7_ ∩ *X*_13_	0.301	NE	*X*_7_ ∩ *X*_18_	0.390	NE	*X*_16_ ∩ *X*_18_	0.457	NE
	*X*_7_ ∩ *X*_11_	0.228	NE	*X*_7_ ∩ *X*_17_	0.358	NE	*X*_16_ ∩ *X*_17_	0.370	NE
	*X*_7_ ∩ *X*_16_	0.228	NE	*X*_7_ ∩ *X*_20_	0.291	NE	*X*_16_ ∩ *X*_20_	0.250	BE
2010–2020	*X*_2_ ∩ *X*_6_	0.236	BE	*X*_2_ ∩ *X*_13_	0.266	BE	*X*_2_ ∩ *X*_20_	0.299	BE
	*X*_2_ ∩ *X*_9_	0.227	BE	*X*_2_ ∩ *X*_10_	0.254	NE	*X*_2_ ∩ *X*_18_	0.292	BE
	*X*_2_ ∩ *X*_8_	0.275	NE	*X*_2_ ∩ *X*_11_	0.287	NE	*X*_2_ ∩ *X*_17_	0.265	NE
	*X*_1_ ∩ *X*_6_	0.230	BE	*X*_1_ ∩ *X*_13_	0.232	BE	*X*_1_ ∩ *X*_20_	0.279	BE
	*X*_1_ ∩ *X*_9_	0.230	BE	*X*_1_ ∩ *X*_10_	0.202	BE	*X*_1_ ∩ *X*_18_	0.249	BE
	*X*_1_ ∩ *X*_8_	0.212	NE	*X*_1_ ∩ *X*_11_	0.269	NE	*X*_1_ ∩ *X*_17_	0.261	NE
	*X*_3_ ∩ *X*_6_	0.299	BE	*X*_3_ ∩ *X*_13_	0.287	BE	*X*_3_ ∩ *X*_20_	0.303	NE
	*X*_3_ ∩ *X*_9_	0.244	BE	*X*_3_ ∩ *X*_10_	0.244	NE	*X*_3_ ∩ *X*_18_	0.299	NE
	*X*_3_ ∩ *X*_8_	0.193	NE	*X*_3_ ∩ *X*_11_	0.242	NE	*X*_3_ ∩ *X*_17_	0.220	NE
	*X*_6_ ∩ *X*_13_	0.299	BE	*X*_6_ ∩ *X*_20_	0.248	BE	*X*_13_ ∩ *X*_20_	0.331	BE
	*X*_6_ ∩ *X*_10_	0.292	BE	*X*_6_ ∩ *X*_18_	0.289	BE	*X*_13_ ∩ *X*_18_	0.284	BE
	*X*_6_ ∩ *X*_11_	0.280	BE	*X*_6_ ∩ *X*_17_	0.319	BE	*X*_13_ ∩ *X*_17_	0.234	BE
	*X*_9_ ∩ *X*_13_	0.265	BE	*X*_9_ ∩ *X*_20_	0.285	BE	*X*_10_ ∩ *X*_20_	0.298	BE
	*X*_9_ ∩ *X*_10_	0.236	BE	*X*_9_ ∩ *X*_18_	0.358	NE	*X*_10_ ∩ *X*_18_	0.306	BE
	*X*_9_ ∩ *X*_11_	0.363	NE	*X*_9_ ∩ *X*_17_	0.261	BE	*X*_10_ ∩ *X*_17_	0.329	NE
	*X*_8_ ∩ *X*_13_	0.305	NE	*X*_8_ ∩ *X*_20_	0.293	NE	*X*_11_ ∩ *X*_20_	0.306	BE
	*X*_8_ ∩ *X*_10_	0.290	NE	*X*_8_ ∩ *X*_18_	0.320	NE	*X*_11_ ∩ *X*_18_	0.310	BE
	*X*_8_ ∩ *X*_11_	0.291	NE	*X*_8_ ∩ *X*_17_	0.315	NE	*X*_11_ ∩ *X*_17_	0.264	BE

NE indicates non-linear enhancement, and BE indicates bi-factor enhancement.

## Data Availability

Data presented in this study are available upon request to the corresponding authors.

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
