# Peer review of "Spatio-Temporal Patterns of County Population Shrinkage and Influencing Factors in the North–South Transitional Zone of China"

_ijerph, 2022, doi:10.3390/ijerph192315801_

Round 1

Reviewer 1 Report

Reviewer’s comments on the manuscript by Wu et al. entitled: Spatial-temporal patterns of county population shrinkage and influencing factors in the North-South Transitional Zone of China.

Manuscript ID: ijerph-2023143

November, 2022.

The topic is interesting. However, the paper have several flaws and need to be revised before publication in the Journal. Specific Comments please check the PDF file.

Author Response

Thank you very much for your review of the manuscript. Those comments are very helpful for revising and improving our paper, as well as the important guiding significance to other research. The author fully adopts the opinions of the reviewers and has revised the manuscript. The specific modifications are as follows:

Point 1: Add explanation for selecting data for 2000, 2010, and 2020.

Response 1: We thank the experts for their valuable comments. We provide additional explanations of the 2000, 2010, and 2020 Chinese census data used in this study.(Page 5, Lines 185-188)

Point 2: Add an explanation for the selection of the study area,and note the superscript format.

Response 2: We have added an explanation of why the North-South Transitional Zone of China was chosen as the study area in this study,and modified the superscript format(Page 5, Lines121-128,Lines 133)

Point 3: Need to revise the conclusion of the paper, not the repetition of the analysis of results.

Response3: Thank you for your valuable comments! I have rewritten the conclusion section of this study.(Page 24-25, Lines 729-773)

Point 4: The references of the article cite too many Chinese journals, some should be deleted and replaced with English literature.

Response 4: I removed the Chinese journal literature cited in this study and replaced it with English journal literature.(Page 26-27, Lines 827-852)

Reviewer 2 Report

It is a pleasure to evaluate your manuscript. Please consider my comments to improve the document.

1. Add the reference for the population shrinkage index.
2. Add the reference for the global Moran index.
3. Please add the references for three population censuses.
4. Please add to the discussion the theoretical implications and managerial implications.
5. It is necessary that the discussion can compare the outcomes with shrinkage in other regions.

Author Response

Thank you very much for your review of the manuscript. Those comments are very helpful for revising and improving our paper, as well as the important guiding significance to other research. The author fully adopts the opinions of the reviewers and has revised the manuscript. The specific modifications are as follows:

Point 1: Add a reference to the population shrinkage index.

Response 1: We thank the experts for your valuable comments, and We added a reference to the population contraction index.(Page 5 line 145)

Point 2: Add a reference to the Global Moran Index.

Response 2: We thank the experts for your valuable comments, and We have added a reference to the Global Moran Index.(Page 5 line 153)

Point 3: Please add references to the three censuses.

Response 3: Thank you for your pertinent advice.We provide additional explanations of the 2000, 2010, and 2020 Chinese census data used in this study.(Page 5, Lines 185-188)

Point 4: Please add theoretical implications and managerial implications to the discussion.

Response 4: We thank the experts for your valuable comments, and we have rewritten the discussion section to add theoretical implications and managerial implications of this study to the discussion and to provide targeted policy recommendations for the region.(Page 22-23, Lines 632-703)

Point 5: Discussion of the need to compare the results with contractions in other regions

Response 5: We thank the experts for your valuable comments, and we compared our discussion with the phenomenon of population contraction in other less developed regions of the world.(Page 22-23, Lines 632-721)

Reviewer 3 Report

This study is very interesting. For a fixed physical geographical area, the landform and climate conditions should be stable in a short period of time, so these indicators are meaningless for this research. According to the research progress in this field, economic and educational indicators are the main driving factors, especially household income, per capita disposable income, education level, and so on, which should be supplemented. At the same time, you had better optimize and adjust the current indicators.

Author Response

Thank you very much for your review of the manuscript. Those comments are very helpful for revising and improving our paper, as well as the important guiding significance to other research. The author fully adopts the opinions of the reviewers and has revised the manuscript. The specific modifications are as follows:

Point 1: For a fixed physical geographic area, topographic and climatic conditions should remain stable over a short period of time, so these indicators are not relevant for this study.

Response 1: We thank the experts for their valuable comments.For the North-South transition zone in China, the landscape and climatic conditions remain stable for a short period of time.I strongly agree with you on this aspect, so we removed the average annual temperature and average annual precipitation metrics. However, the average altitude and topographic relief are retained because the topography of the North-South Transitional Zone in China, which is high in the west and low in the east, leads to a clear geographical differentiation in the level of regional economic and social development, which also has an important impact on the regional population shrinkage.(Page 12 Line 372)

Point 2: According to the research progress in this field, economic and educational indicators are the main drivers, especially household income, disposable income per capita, and education level, and should be supplemented. Current indicators need to be optimized and adjusted.

Response 1: According to the progress of existing research on population contraction, we have adjusted and optimized the indicator system of this study, and added new disposable income per capita in terms of economic factors, and education level already existed in the original indicator system. Unfortunately, the indicator of household income was not added to the indicator system of this study because the data were not available. Finally, a total of 20 impact factors were selected for the optimized indicator system, and the corresponding manuscript contents were rewritten.(Page 11-21 Lines 348-628)

Reviewer 4 Report

The article addresses the current problem of population shrinkage. The authors have made an exhaustive analysis in the context of trends in relation to the formulated problem and the factors affecting it. The authors rightly note that the analysis carried out can be used in the future to effectively coordinate the movement of population and ensure the sustainable development of regions. In my opinion, the research presented in the article is neither new nor groundbreaking, but it is important because of its applicability to the management of regions prone to shrinkage.

I have no substantive comments to the content of the article. I only indicate that on page 5 (line 173) it should be Table 1.

Author Response

Thank you very much for your review of the manuscript. Those comments are very helpful for revising and improving our paper, as well as the important guiding significance to other research. The author fully adopts the opinions of the reviewers and has revised the manuscript. The specific modifications are as follows:

Point 1: I have no substantive comments to the content of the article. I only indicate that on page 5 (line 173) it should be Table 1.

Response 1: We thank the experts for their valuable comments, and we have made changes in our manuscript (page 5 line 180).We also double-checked all titles, figure names, and table names in the manuscript to ensure that they are correct.

Round 2

Reviewer 1 Report

The authors have been revisied the manuscript content according to my suggestion, I think can be published.

Reviewer 3 Report

 Please check the text and data carefully and correct the mistakes.